# Longitudinal Change of Forearm-Hand Inertia Value and Shoulder Musculature Using Dual X-ray Absorptiometry in Youth Japanese Baseball Players: Implications for Elbow Injury

**DOI:** 10.3390/sports8120152

**Published:** 2020-11-25

**Authors:** Toshiharu Tsutsui, Toshihiro Maemichi, Satoshi Iizuka, Suguru Torii

**Affiliations:** 1Graduate School of Sport Sciences, Waseda University, 2-579-15 Mikajima, Tokorozawa, Saitama 359-1192, Japan; t.m.waseda@ruri.waseda.jp; 2Japan Institute of Sports Science, 3-15-1 Nishigaoka, Kita-ku, Tokyo 115-0056, Japan; anporian@gmail.com; 3Faculty of Sport Sciences, Waseda University, 2-579-15 Mikajima, Tokorozawa, Saitama 359-1192, Japan; shunto@y.waseda.jp

**Keywords:** peak height velocity, DXA, adolescent, elbow injury

## Abstract

It is important to understand the timing of the maximum increase of forearm-hand inertia value and lean body mass (LBM) of the shoulder girdle musculature when elbow injury frequently occurs. This study aimed to clarify the discrepancies of those in youth baseball players. Forty-three male baseball players (8- to 14-years-old) participated in this study. The forearm-hand inertia value and LBM of the shoulder girdle musculature were calculated using dual-energy X-ray absorptiometry (DXA). A cubic spline fit was applied to the annual increase forearm-hand inertia value and LBM of the shoulder girdle musculature for each chronological age and years from peak height velocity (PHV) age. As a result of cubic splines fitting, the peak timing for forearm-hand inertia value and LBM of the shoulder girdle musculature was 12.42 and 12.75 years in chronological age, −0.66 and −0.11 years in PHV age. Therefore, the peak timing of forearm-hand inertia value was about 4 months earlier in chronological age and half a year earlier in PHV age than LBM of the shoulder girdle musculature. Acquiring sufficient shoulder girdle musculature during the period when the growth of the shoulder girdle musculature cannot catch up with forearm-hand inertial value is necessary to reduce the elbow load while throwing.

## 1. Introduction

Baseball is a popular sport all over the world, and many children start playing it at an early age. As an injury specific to youth baseball players, elbow injury is reported to occur in approximately a quarter of the pitchers aged between 9 and 12 years [1,2]. The suggested risk factors for elbow injury in youth baseball players are mostly associated with pitching. These risk factors include the number and frequency [3,4], position [1,5], bad posture, flexibility of the lower limb muscles [6], and poor throwing biomechanics, like lowering the position of the elbow and using only the elbow to the periphery [7].

It has been reported that the growth pattern of each joint segmentation of the extremities proceeds peripherally during the youth stage [8]. Moreover, as body proportions including adult and youth mass ratios have been shown to differ, the ratio of length and mass in each segment varies through the growth period. According to a report investigating the relationship between the elbow varus torque during throwing and the segment mass of the upper limb by using dual-energy X-ray absorptiometry (DXA), the elbow varus torque has a positive relationship with the mass of the hand and forearm [9]. Accordingly, physical characteristics which growth peripherally peculiar to the growth period may influence elbow injury. However, the above study did not consider that the segment center of gravity is far from the body center because the investigation was based on the mass of only three segments: the hand, forearm, and upper arm. Therefore, adapting the estimation of inertial value [10] is essential to reflect the characteristics of physical growth. Since the inertia value is multiplied by the increment of the distance of each mass, it is estimated that the influence of growth for each segment becomes even greater. In other words, by calculating the inertia value distal to the elbow joint (forearm-hand inertia value), it is possible to identify the timing at which elbow injury is likely to occur from the aspect of physical growth characteristics.

Meanwhile, the shoulder girdle musculature exerts high muscle activity from the arm-cocking to the arm-acceleration phases in throwing, which is exposed to the highest mechanical load of the elbow joint [11]. Hence, the shoulder girdle musculature is expected to play a significant role in supporting the upper limb on the pitching side while throwing. Accordingly, it is important to understand the timing of the maximum increase of forearm-hand inertia value and the lean body mass (LBM) of the shoulder girdle musculature during the ages at which elbow injury frequently occurs. Therefore, this study had two purposes: First, to clarify the timing of maximum increase in forearm-hand inertia value; and second, to investigate the discrepancy in the timing of a maximum increase in forearm-hand inertia value and LBM of the shoulder girdle musculature in youth baseball players. We hypothesized that the forearm-hand inertia value would precede LBM of the shoulder girdle musculature because the growth pattern of each segment for the extremities proceeds peripherally during youth. In addition, as an indicator of growth, peak height velocity (PHV) age which shows the growth spurt and establishes well as a marker was also analyzed since the timing of growth would differ depending on individual even if they were in the same age.

## 2. Materials and Methods

### 2.1. Participants

Forty-three healthy Japanese male baseball players, aged 8–14 years, who took part in the second measurements one year after the first participated in this study. They did not perform any specific strength training and exercises regarding the upper limbs. The participants attended regular baseball practice and games on the weekends for more than 3 h. They were all in good health and free from disorders affecting the growth of their extremities and body. In addition, we confirmed that they had baseball experience of more than 1 year. Since the participants were minors, we explained the intent of this study to their parents and obtained their consent before the experiment. The ethics committee of the Faculty of Sport Sciences, Waseda University (2017-323), approved this study.

### 2.2. Procedure and Data Collection

First, we collected the dates of birth of the participants to calculate the chronological age. After that, anthropometric measurements including height and weight were taken with light clothing. The height was measured without shoes, using a stadiometer to the nearest 0.1 cm. Then, weight was measured without shoes, using a digital scale to the nearest 0.1 kg. After the anthropometric measurements, DXA measurement was performed.

Whole body DXA scans were performed within the Waseda University using a Hologic QDR densitometer and analyzed with version 12.4.3 software (Hologic, Bedford, MA, USA) after calibration by qualified personnel (S.T.). The participants were placed in the supine position such that the midline of the body coincided with the centerline of the examination table. The supine posture involved the following: the shoulder joint at 45°, forearm pronated, hip and knee fully extended, and feet fixed in way that the toes were in contact and did not move during scanning. Sub-Region, the analysis mode of DXA, was used for data analysis. We acquired the data compartmentalized to the smallest and arranged at constant intervals from the elbow joint that passes through the lateral epicondyle of the humerus to the end of the fingers on the pitching side (Figure 1A). Then, based on the procedure for calculating the moment of inertia reported by Ganley and Powers [10], the integrated value of the square of the mass in each section and the distance from the center of the elbow joint was calculated as the forearm-hand inertia value. In addition, the region of interest for estimating the LBM of the shoulder girdle supporting the upper limbs was compartmentalized using the Sub-Region mode. At this time, the section of the shoulder girdle was compartmentalized as below (Figure 1B): soft tissue margin of the shoulder (upper edge), the lower edge of 12th thoracic vertebra (lower edge), outside line of the thoracic spine (inside edge), and soft tissue margin of the chest (outside edge). This section includes the serratus anterior, rhomboid, trapezius, pectoralis major, levator scapula, rotator cuff, deltoid, and latissimus dorsi muscles, which are often used to measure muscle activity during pitching [11].

The PHV age of each participant was evaluated using the AUXAL 3.1 program (Scientific Software International Inc., Skokie, IL, USA), using their past height records from six years old and their height measurement in this study. We collected the data regarding their heights, measured at school annually during elementary school from their parents. We calculated the years from PHV age of each participant by subtracting the chronological age from PHV age to determine the period until the peak of growth. In other words, it means before the peak of growth if years from PHV age is negative, or before the peak of growth if years from PHV age is positive.

### 2.3. Statistics

Descriptive statistics (means ± standard deviation and 95% confident intervals) are reported about PHV age, baseball experience in initial measurement, and chronological age, height, weight, years from PHV, forearm-hand inertia value, and LBM of the shoulder girdle musculature regarding the measurement of both initial and one-year later. Then, a paired samples *t*-test was performed to examine the differences between initial and one-year later measurements. In addition, a cubic spline fit was applied to the annual increase in forearm-hand inertia value and LBM of the shoulder girdle musculature for each chronological age and year from PHV at the midpoint of the initial and 1-year-later measurement; this determined the difference in peak timing between these two parameters, generated from the entire pooled data. Statistical analyses were performed using SPSS for Windows (IBM SPSS version 24.0; SPSS Inc., Chicago, IL, USA). A *p*-value < 0.05 was used to determine statistical significance.

## 3. Results

Table 1 shows descriptive data for PHV age, baseball experience, age, height, weight, years from PHV, forearm-hand inertia value, and LBM of the shoulder girdle musculature regarding the initial and 1-year-later measurements. The measurements taken after 1 year showed higher values than initial measurement in all parameters.

Figure 2 shows the relationship between chronological age and forearm-hand inertia value and LBM of the shoulder girdle musculature. A cubic spline of the annual increase in both forearm-hand inertia values (R^2^ = 0.358, *p* < 0.001) and that of LBM of the shoulder girdle musculature (R^2^ = 0.765, *p* < 0.001) are well fitted. In addition, the peak timing was 12.42 years for forearm-hand inertia value and 12.75 years for LBM of the shoulder girdle musculature.

Figure 3 shows the relationship between years from PHV age and forearm-hand inertia value and LBM of the shoulder girdle musculature. A cubic spline of the annual increase in forearm-hand inertia values (R^2^ = 0.540, *p* < 0.001) and LBM of the shoulder girdle musculature (R^2^ = 0.632, *p* < 0.001) were well fitted. The peak timing was −0.66 years for forearm-hand inertia value and −0.11 years for LBM of the shoulder girdle musculature.

## 4. Discussion

This study aimed to clarify the timing of the maximum increase in forearm-hand inertia and LBM of the shoulder girdle musculature and their discrepancy in youth baseball players. This is the only study that shows the implication for elbow injury by calculating forearm-hand inertia value from DXA and that investigates the aspect of physical growth longitudinally based on the characteristics of the youth stage when preceding growth from the periphery.

As a result of the examination of chronological age and the timing of a maximum increase in forearm-hand inertia value and LBM of the shoulder girdle (Figure 2), the peak timing of forearm-hand inertia value (12.42 years) is about 4 months earlier than LBM of the shoulder girdle (12.75 years). Fleisig et al. [7] clarified that elbow varus torque would increase greatly from 13 years old and guessed the coincidence with physical advances after puberty. Lyman et al. [12] have reported that the risk factor for elbow injury in baseball players aged 9–12 was about three times the odds ratio, especially at age 12 years and above. Since the timing of the maximum increase in forearm-hand inertia values in this study coincided with the age at which elbow injury frequently occurs, it can be concluded that the physical growth characteristics during the youth stage may be involved in the risk of elbow injury.

Meanwhile, the result of the relationship between PHV age and the timing of the maximum increase in the forearm-hand inertia value and LBM of the shoulder girdle identified that the peak timing of forearm-hand inertia value (−0.66 years) is about half a year earlier compared to lean mass of the shoulder girdle (−0.11 years). Given that the mean PHV age among the participants was 13.2 ± 0.6 years, no significant difference was seen compared to the age of 13.1 ± 1.0, reported as the PHV age of Japanese baseball players [13]. A study showing longitudinal changes of LBM in Japanese adolescents reported that the trunk and upper limb LBM peaked 0.08 and 0.16 years earlier than PHV age, respectively [14]. The timing of the maximum increase in LBM of the shoulder girdle musculature in this study was 0.11 years earlier than PHV age, which was within the period shown in the previous study, and therefore supported the result of the current study. In addition, it is predicted that the LBM of the shoulder girdle musculature, which supports the throwing upper limb, is temporarily insufficient for about half a year, the term of the discrepancy. The shoulder girdle musculature is essential for supporting the upper limbs from the arm-cocking phase to the arm-acceleration phase [11] and contributes to avoiding the local stress on the elbow. It is necessary to acquire sufficient shoulder girdle musculature, especially during the period when the growth of the shoulder girdle musculature cannot catch up with the forearm-hand inertial value that peaks after age 12 and half a year before PHV age.

Shanley et al. [15] have classified the risk factors as nonmodifiable and modifiable. Modifiable risk factors refer to pitching, including number and frequency [1,2], physical functions like flexibility of the lower limb muscles [6], and posture [5]. Nonmodifiable risk factors include age and height [1]. The inertia value calculated in this study is an indicator that cannot be changed because it reflects the physical growth characteristics. Therefore, it is necessary for the coaches, trainers, and other instructors involved with youth baseball players to understand when the forearm-hand inertia value reaches its peak. Taking into account modifiable risk factors, it is important to increase the elbow joint flexion angle for reducing the valgus torque during pitching because youth baseball players have less elbow flexion angle [16] when throwing compared to high school, college, and professional [17]. Furthermore, since the weight of the ball (142 g) is added to forearm-hand inertial value at the end of the upper limb during actual pitching, the adaptation of a light ball (113 g), as reported by Fleisig et al. [18], might help in reducing elbow load.

This research has two limitations. First, we did not investigate the relationship between forearm-hand inertia values and actual elbow injury. Second, the LBM of the shoulder girdle musculature is not just a muscle mass because it includes internal organs as lean mass in DXA measurement, although Midorikawa et al. [19] found that DXA derived prediction equations are accurate for the estimation of skeletal muscle mass and Bridge et al. [20] reported that the coefficient of concordance between the DXA and magnetic resonance imaging is significant for monitoring the longitudinal change of lean soft tissue mass during periods of growth.

## 5. Conclusions

The peak timing of forearm-hand inertia value (12.42 years) is about 4 months earlier than the LBM of the shoulder girdle musculature (12.75 years) in chronological age. Moreover, the peak timing of forearm-hand inertia value (−0.66 years) is about half a year earlier than the LBM of the shoulder girdle musculature (−0.11 years) in PHV age. Therefore, it is necessary to acquire sufficient shoulder girdle musculature during the period when the growth of the shoulder girdle musculature cannot catch up with forearm-hand inertial value.

## Figures and Tables

**Figure 1 sports-08-00152-f001:**
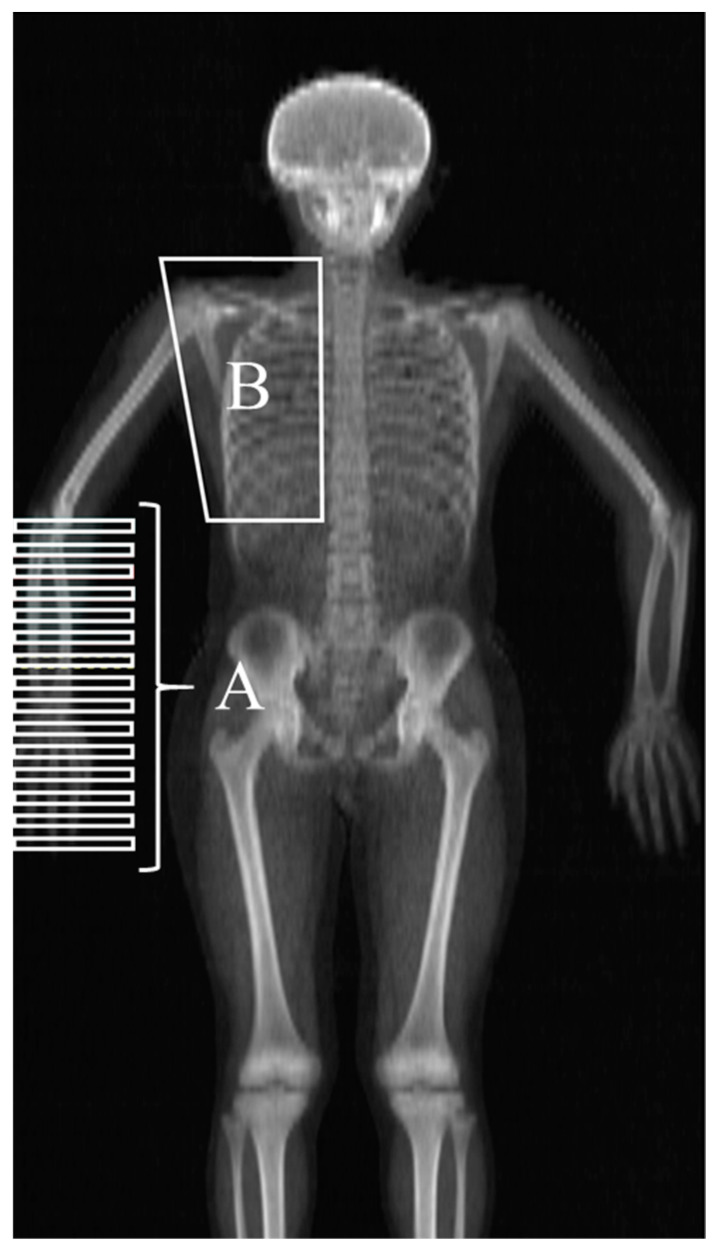
Compartment of Sub-Region analysis in a typical image of a whole body dual-energy X-ray absorptiometry (DXA) scan. (A) Sections arranged from elbow joint to the end of the finger. (B) Shoulder girdle compartment consisted of 4 lines; soft tissue margin of shoulder (Upper Edge), lower edge of 12th thoracic vertebra (Lower Edge), outside line of the spine (Inside Edge), soft tissue margin of chest (Outside Edge).

**Figure 2 sports-08-00152-f002:**
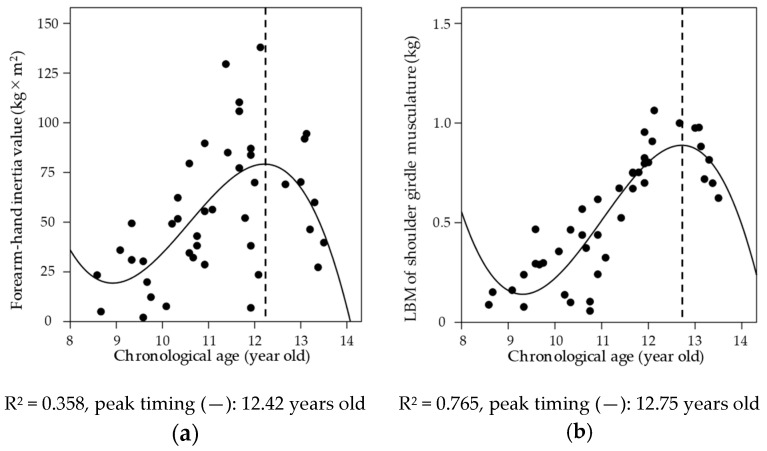
The relationships between years from chronological age and (**a**) forearm-hand, inertia value, and (**b**) lean body mass (LBM) of the shoulder girdle musculature.

**Figure 3 sports-08-00152-f003:**
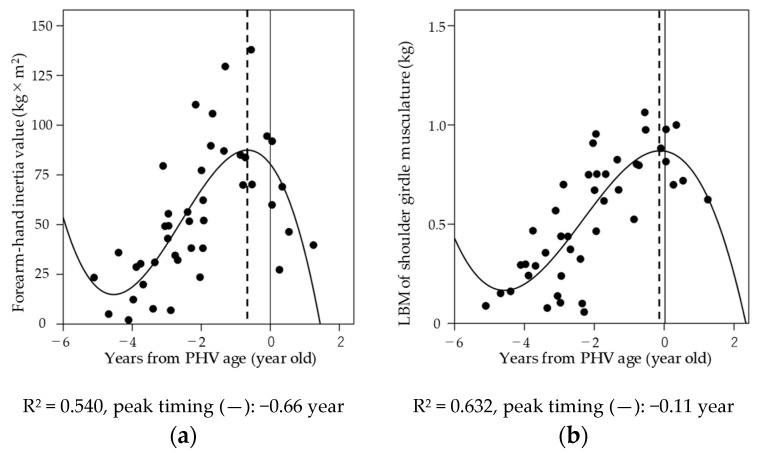
The relationships between years from peak height velocity (PHV) age and (**a**) forearm-hand, inertia value, and (**b**) LBM of the shoulder girdle musculature.

**Table 1 sports-08-00152-t001:** Participant characteristics.

Variables	Mean ± SD (95%CI)				
PHV age (years)	13.2 ± 0.6		(13.1–13.4)		
Baseball Experience	3.1 ± 1.7		(2.2–3.9)		
(years)					
	Initial		One year later		*p*
Chronological age (years old)	11.1 ± 1.1	(10.7–11.6)	12.1 ± 1.1	(11.7–12.6)	<0.001
Height (cm)	143.6 ± 9.9	(139.4–147.8)	151.4 ± 10.7	(146.7–155.9)	<0.001
Weight (kg)	38.6 ± 12.5	(33.4–43.9)	44.5 ± 13.6	(38.8–50.3)	<0.001
Years from PHV age (years)	−2.2 ± 1.4	(−2.8–−1.6)	−1.2 ± 1.3	(−1.8–−0.6)	<0.001
Forearm-hand inertia value (kg × m^2^)	204.0 ± 90.1	(165.9–242.0)	270.3 ± 92.2	(226.1–314.5)	<0.001
LBM of shoulder girdle musculature (kg)	2.3 ± 1.0	(1.8–2.7)	2.9 ± 1.1	(2.5–3.4)	<0.001

(SD: standard deviation, 95%CI: 95% confidence interval).

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
