# Peer review of "Longitudinal Change of Forearm-Hand Inertia Value and Shoulder Musculature Using Dual X-ray Absorptiometry in Youth Japanese Baseball Players: Implications for Elbow Injury"

_sports, 2020, doi:10.3390/sports8120152_

Round 1

Reviewer 1 Report

This manuscript reports an interesting study where the purpose was to examine the longitudinal change of forearm-hand inertia value and shoulder muscle in young baseball players. It was thought that information on this topic could have implications for elbow injury in these young throwing athletes. A total of 43 male baseball players between 8 and 14 years of age participated in the study. The timing of the maximum increase in forearm-hand inertia and lean body mass of the shoulder were quantified and cubic spline fitting of the annual increases (initial and 1 year later measurements) was used to determine the age at which the above two variables occurred.

The main findings were: 1) the peak timing of forearm-hand inertia was 12.41; 2) the peak value of shoulder girdle lean body mass was 12.75 years; and 3) obviously this resulted in a 4 month time lag between the two measures. Therefore, the authors speculated that methods to increase shoulder girdle muscle mass during these time periods should be emphasized as it would be assumed this could prevent injury.

The study is novel and the results have practical applications. Many aspects of the methods appear to have been done in an appropriate manner.

Overall, I have 1 major comments and a few minor comments that the authors should address.

Major:

  1. The measure of shoulder girdle muscle mass is problematic and at the least needs more details provided. What are all the muscles involved in this measure? IThe authors do say that internal organs are included and give a reference as to why this is also okay. Are there any other studies that can make the readers confident that the measure of muscle mass was valid and reliable based on the two above comments?

Minor:

  1. There are a few issues with the English throughout the text. The authors did a great job with the English as I am sure it is their second language and the mistakes are very minor. Nonetheless, there are a pretty high amount of small grammatical mistakes in the paper. It is strongly suggested that the authors have an English speaking proofreader correct these minor errors.

As one example on lines 50 and 52 “shoulder girdle muscle” should read “shoulder girdle muscles” of “shoulder girdle musculature”

  1. Line 118 “The measurements taken after 1 year showed high values in all parameters”. Do the authors mean to say all the measurements increased over time substantially? Please reword.

Author Response

Responses to the reviewers’ comments

We wish to express our appreciation to the reviewers for their insightful comments on our paper. We responded to each of the reviewers’ comments and suggestions below. Revisions in the text are indicated by yellow highlighting.

【Response to Reviewer 1】

Thank you very much for providing important comments. We are thankful for the time and energy you expended. Our responses to the referees’ comments are as follow:

Reviewer Major Comment 1:

The measure of shoulder girdle muscle mass is problematic and at the least needs more details provided. What are all the muscles involved in this measure? The authors do say that internal organs are included and give a reference as to why this is also okay. Are there any other studies that can make the readers confident that the measure of muscle mass was valid and reliable based on the two above comments?

I added the explanation about the detail of shoulder muscle which partitioned as shoulder girdle musculature in line98-100; This section includes the serratus anterior, rhomboid, trapezius, pectoralis major, levator scapula, rotator cuff, deltoid and latissimus dorsi muscles, which are often used to measure muscle activity during pitching [11]. In addition, to support the explanation of the accuracy of the measurement, I inserted a reference that was comparing lean soft tissue mass with MRI on lines 195-197; Bridge et al. [20] reported that the coefficient of concordance between the DXA and magnetic resonance imaging is significant for monitoring the longitudinal change of lean soft tissue mass during periods of growth.

Reviewer Minor Comment1: 

There are a few issues with the English throughout the text. The authors did a great job with the English as I am sure it is their second language and the mistakes are very minor. Nonetheless, there are a pretty high amount of small grammatical mistakes in the paper. It is strongly suggested that the authors have an English speaking proofreader correct these minor errors.

As one example on lines 50 and 52 “shoulder girdle muscle” should read “shoulder girdle muscles” of “shoulder girdle musculature”

I revised the title, summary and the article as you pointed out. There are 19 corrections in total. The line numbers of the corrected parts are shown below.

■Title (Line3)

■Abstract (Line14, 17, 19, 21, 24 and 25)

■Article

  1. Introduction (Line 52, 54, 57, 60 and 62)
  2. Materials and Methods (Line 111 and 115)
  3. Results (Line 127, 131, 132, 134, 136, 137, 139, 143, 145 and Figure 2 and 3)
  4. Discussion (Line 148, 169, 172, 173, 176 and 192)
  5. Conclusions (Line 200, 202 and 203)

Line 118 “The measurements taken after 1 year showed high values in all parameters”. Do the authors mean to say all the measurements increased over time substantially? Please reword.

I used the expression that I intended that the measurement results after one year were all improved due to the influence of physical growth. I performed a comparison between initial and one year later measurement (a paired t-test; line 112-113, 128-129) and updated Table1.

Reviewer 2 Report

Well done on the following manuscript. Please find my comments below.

General:

Considering it is a limitation of the study that it did not investigate the relationship between hand values and elbow injury, there needs to be more evidence to support this assumption. There is a comment in the introduction that it is possible, but I feel this assumption needs to be backed up with more evidence.

Aspects of the methods are really good, however, I felt there was some important information missing. For example, there was no comment about when the initial and then post- analysis was actually completed. I only worked this out from the table in the results section. 

Specific:

Line 32: Delete 2nd the: “The suggested the risk”

Line 32-35: The following sentence needs to be re-worded. “The suggested the risk factors for elbow injury in youth baseball players are pitching, including the number and frequency [3,4], position [1,5], bad posture, flexibility of the lower limb muscles [6], and poor throwing biomechanics, like lowering the position of the elbow and using only the elbow to the periphery [7].”

Maybe break into two sentences commencing with “The suggested risk factors for elbow injury in youth baseball are mostly associated with pitching. These risk factors include…”

Line 36-38: Be more specific as to the major segment you are discussing.

Line 42: What are you trying to say here- “Accordingly, physical characteristics peculiar to the growth period may influence elbow injury.”

I assume you are trying to say an abnormal growth rate (larger or smaller) may lead to an increase in injury? This needs to be re-worded and made clearer.

Line 43-49: This section really highlights the difference from the previous study and highlights why the present investigation is important.

Line 65-66: What is meant by “who did not perform any specific training and exercises participated in this study.”

This needs to be made clear.

Methods

Line 93: Repeating that you have collected height and weight. I would suggest adding age being collected with the earlier explanation and then this could be deleted.

Procedures

It is not until table 1 that I realise you took the second measurements one year after the first. This must be made very clear within the methods section.

Results

Table 1: I really think it would be worthwhile showing the comparison from the initial to post-1 year testing results. Can the table be adapted to show a percentage change or some other comparison between the variables?

Author Response

Responses to the reviewers’ comments

We wish to express our appreciation to the reviewers for their insightful comments on our paper. We responded to each of the reviewers’ comments and suggestions below. Revisions in the text are indicated by yellow highlighting.

【Response to Reviewer 2】

Thank you very much for providing important comments. We are thankful for the time and energy you expended. Our responses to the referees’ comments are as follow:

General:

Considering it is a limitation of the study that it did not investigate the relationship between hand values and elbow injury, there needs to be more evidence to support this assumption. There is a comment in the introduction that it is possible, but I feel this assumption needs to be backed up with more evidence.

Previous studies have shown that there is a positive association between forearm and hand weight and elbow valgus torque generated during pitching ([reference 9]; line 40-43). Therefore, we believe that there is a great possibility that it is related to elbow injuries in youth period. The originality of this research is to make the segments finer, and we believe that the implications will contribute to future studies which include.

Aspects of the methods are really good, however, I felt there was some important information missing. For example, there was no comment about when the initial and then post- analysis was actually completed. I only worked this out from the table in the results section. 

I added the sentences in Material and Method section about measurement timing. I revised to “Forty-three healthy Japanese male baseball players, aged 8-14 years, who took part in the second measurements one year after the first participated in this study.” (line 68-69).

Specific:

Line 32: Delete 2nd the: “The suggested the risk”

I deleted 2nd “the” and revised “The suggested risk factors” in line 33.

Line 32-35: The following sentence needs to be re-worded. “The suggested the risk factors for elbow injury in youth baseball players are pitching, including the number and frequency [3,4], position [1,5], bad posture, flexibility of the lower limb muscles [6], and poor throwing biomechanics, like lowering the position of the elbow and using only the elbow to the periphery [7].”

Maybe break into two sentences commencing with “The suggested risk factors for elbow injury in youth baseball are mostly associated with pitching. These risk factors include…”

I revised from “The suggested risk factors for elbow injury in youth baseball players are pitching, including the number and frequency [3,4], position [1,5], bad posture, flexibility of the lower limb muscles [6], and poor throwing biomechanics, like lowering the position of the elbow and using only the elbow to the periphery [7].” to “The suggested risk factors for elbow injury in youth baseball players are mostly associated with pitching. These risk factors include the number and frequency [3,4], position [1,5], bad posture, flexibility of the lower limb muscles [6], and poor throwing biomechanics, like lowering the position of the elbow and using only the elbow to the periphery [7].” (line 33). 

Line 36-38: Be more specific as to the major segment you are discussing.

I revised from “each segment” to “each joint segmentation” in line 37.

Line 42: What are you trying to say here- “Accordingly, physical characteristics peculiar to the growth period may influence elbow injury.”

I assume you are trying to say an abnormal growth rate (larger or smaller) may lead to an increase in injury? This needs to be re-worded and made clearer.

I added the explanation of “which growth peripherally” into that sentences. I revised it like “Accordingly, physical characteristics which growth peripherally peculiar to the growth period” in line 43.

Line 43-49: This section really highlights the difference from the previous study and highlights why the present investigation is important.

I revised from “adapting the estimation of inertial value [10] is expected to be useful” to “adapting inertial value estimates is essential to reflect the characteristics of physical growth” to emphasize the highlights (line 46-47).

Line 65-66: What is meant by “who did not perform any specific training and exercises participated in this study.”

This needs to be made clear.

I revised from “who did not perform any specific training and exercises participated in this study” to “who did not perform any specific strength training and exercises regarding the upper limbs participated in this study” (line 68).

Methods

Line 93: Repeating that you have collected height and weight. I would suggest adding age being collected with the earlier explanation and then this could be deleted.

I deleted the sentence “Participant characteristics, including age, height, and weight, were collected.” Then, I added the information of age into 1st paragraph of 2.2. Procedure and Data Collection: “First, we collected the dates of birth of the participants to calculate the chronological age.” (line 77)

Procedures

It is not until table 1 that I realise you took the second measurements one year after the first. This must be made very clear within the methods section.

I added the explanations about the number of measurements in line 68-69 like “who took part in the second measurements one year after the first participated in this study

Results

Table 1: I really think it would be worthwhile showing the comparison from the initial to post-1 year testing results. Can the table be adapted to show a percentage change or some other comparison between the variables?

I revised and performed a comparison between initial and one year later measurement (a paired t-test; line 112-113, 128-129) and updated Table1.

Reviewer 3 Report

The present study aimed to clarify the discrepancies of those in youth baseball players. The authors concluded that to acquire sufficient shoulder girdle muscle during the period when the growth of shoulder girdle muscle cannot catch up with forearm-hand inertial value is necessary to reduce the elbow load while throwing. Generally speaking, the article is interesting and clinically important. Before acceptance, I would like to make some comments for the authors to follow.

  1. Line 30: Baseball is a popular sport all over the world. The authors do not need to specify Japan.
  2. Line 37: What is the scaled mass ratio? Please clarify.
  3. Please use some subtitles in the method portion to make the reader read easier.
  4. In the study, the authors did not use to ultrasound to scan whether the participants have relevant musculoskeletal injury. I would suggest the authors to list this as a limitation and reference the following two articles: Ultrasound measurements of superficial and deep masticatory muscles in various postures: reliability and influencers. Sci Rep. 2020; Utility of sonoelastography for the evaluation of rotator cuff tendon and pertinent disorders: a systematic review and meta-analysis. Eur Radiol. 2020.
  5. Regarding Table 1, please check whether the values denote standard errors or standard deviations. For example, I found the stand deviation of the PHV age is very small. It is likely to be the standard error. Please kindly check.

Author Response

Responses to the reviewers’ comments

We wish to express our appreciation to the reviewers for their insightful comments on our paper. We responded to each of the reviewers’ comments and suggestions below. Revisions in the text are indicated by yellow highlighting.

【Response to Reviewer 3】

Thank you very much for providing important comments. We are thankful for the time and energy you expended. Our responses to the referees’ comments are as follow:

The present study aimed to clarify the discrepancies of those in youth baseball players. The authors concluded that to acquire sufficient shoulder girdle muscle during the period when the growth of shoulder girdle muscle cannot catch up with forearm-hand inertial value is necessary to reduce the elbow load while throwing. Generally speaking, the article is interesting and clinically important. Before acceptance, I would like to make some comments for the authors to follow.

  1. Line 30: Baseball is a popular sport all over the world. The authors do not need to specify Japan.

I revised the sentence from “in Japan” to “all over the world” (line 30).

  1. Line 37: What is the scaled mass ratio? Please clarify.

I revised the sentence from “adult and youth scaled mass ratios” to “body proportions including adult and youth mass ratios” (line 38-39).

  1. Please use some subtitles in the method portion to make the reader read easier.

I added three subtitles in Materials and Methods: 2.1. Participants; 2.2. Procedure and Data Collection; 2.3. Statistics

  1. In the study, the authors did not use to ultrasound to scan whether the participants have relevant musculoskeletal injury. I would suggest the authors to list this as a limitation and reference the following two articles: Ultrasound measurements of superficial and deep masticatory muscles in various postures: reliability and influencers. Sci Rep. 2020; Utility of sonoelastography for the evaluation of rotator cuff tendon and pertinent disorders: a systematic review and meta-analysis. Eur Radiol. 2020.

As you said, this study has not confirmed the relationship with elbow injuries, and we have not performed diagnostic imaging. It is just an implication from the aspect of physical growth. In the future, we plan to investigate the relationship with injuries in detail, so we would like to refer to it in that case.

  1. Regarding Table 1, please check whether the values denote standard errors or standard deviations. For example, I found the stand deviation of the PHV age is very small. It is likely to be the standard error. Please kindly check.

The numbers shown in Table 1 were correct. In particular, PHV age would be said to occur around the age of 12 to 14. As you said, the range of PHV age targeted in this study was relatively small, but there was no mistake.

Round 2

Reviewer 2 Report

I am happy that the author's have addressed my concerns, or, provided clarification to my points.